# Sparse multitask group Lasso for genome-wide association studies

**Asma Nouira** [1,2,3,4]*, **Chloé-Agathe Azencott**[1,2,3]

**1** Mines ParisTech, PSL Research University, CBIO-Centre for Computational Biology, Paris, France, **2** Institut Curie, PSL Research University, Paris, France, **3** INSERM, U1331, Paris, France, **4** UR7537 BioSTM, Biostatistique, Traitement et Modélisation des Données Biologiques, Université Paris Cité, Paris, France

\* asma.nouira.91@gmail.com

## Abstract

A critical hurdle in Genome-Wide Association Studies (GWAS) involves population stratification, wherein differences in allele frequencies among subpopulations within samples are influenced by distinct ancestry. This stratification implies that risk variants may be distinct across populations with different allele frequencies. This study introduces Sparse Multitask Group Lasso (SMuGLasso) to tackle this challenge. SMuGLasso is based on MuGLasso, which formulates this problem using a multitask group lasso framework in which tasks are subpopulations, and groups are population-specific Linkage-Disequilibrium (LD)-groups of strongly correlated Single Nucleotide Polymorphisms (SNPs). The novelty in SMuGLasso is the incorporation of an additional $\ell_1$-norm regularization for the selection of population-specific genetic variants. As MuGLasso, SMuGLasso uses a stability selection procedure to improve robustness and gap-safe screening rules for computational efficiency. We evaluate MuGLasso and SMuGLasso on simulated data sets as well as on a case-control breast cancer data set and a quantitative GWAS in *Arabidopsis thaliana*. We show that SMuGLasso is well suited to addressing linkage disequilibrium and population stratification in GWAS data, and show the superiority of SMuGLasso over MuGLasso in identifying population-specific SNPs. On real data, we confirm the relevance of the identified loci through pathway and network analysis, and observe that the findings of SMuGLasso are more consistent with the literature than those of MuGLasso. All in all, SMuGLasso is a promising tool for analyzing GWAS data and furthering our understanding of population-specific biological mechanisms.

## Author summary

Genome-Wide Association Studies (GWAS) scan thousands of genomes to identify loci associated with a complex trait. However, population stratification, which is the presence in the data of multiple subpopulations with differing allele frequencies, can lead to false associations or mask true population-specific associations. We recently

**Data availability statement:** Code is available in: https://github.com/asmanouira/SMuGLasso. The dataset "General Research Use" in DRIVE

Breast Cancer OncoArray Genotypes is available from the dbGaP controlled-access portal, under Study Accession phs001265.v1.p1 (https://www.ncbi.nlm.nih.gov/projects/gap/cgi-bin/study.cgi?study_id=phs001265.v1.p1). Researchers can gain access the data by applying to the data access committee, see https://dbgap.ncbi.nlm.nih.gov.

**Funding:** This work was supported by the French Agence Nationale de la Recherche (ANR-18-CE45-0021-01 and ANR-19-P3IA-0001 to C-A.A.). The funders had no role in study design, data collection and analysis, decision to publish, or preparation of the manuscript.

**Competing interests:** The authors have declared that no competing interests exist.

proposed MuGLasso, a new computational method to address this issue. However, MuGLasso relied on an ad-hoc post-processing of the results to identify population-specific associations. Here, we present SMuGLasso, which directly identifies both global and population-specific associations. We evaluate both MuGLasso and SMuGLasso on several datasets, including both case-control (such as breast cancer vs. controls) and quantitative (for example, plant flowering time) traits, and show on simulations that SMuGLasso is better suited than MuGLasso for the identification of population-specific associations. In addition, SMuGLasso's findings on real case studies are more consistent with the literature than that of MuGLasso, which is possibly due to false discoveries of MuGLasso. These results show that SMuGLasso could be applied to other complex traits to better elucidate the underlying biological mechanisms.

## Introduction

Feature selection methods have emerged as a popular way of framing Genome-Wide Association Studies (GWAS) to uncover the genetic underpinnings of complex diseases, such as cancer. GWAS aim at establishing associations between genetic variants, more specifically Single Nucleotide Polymorphisms (SNPs), and the presence/absence of a disease or a quantitative trait [1–3]. However, their ability to identify relevant variants is limited by several difficulties, including the curse of dimensionality, population stratification, linkage disequilibrium, and the lack of stability of feature selection procedures with respect to small changes in the input samples. Consequently, the application of feature selection requires careful attention to mitigate false discoveries. The challenge in this context is optimizing the stability of selection to identify regions of interest while minimizing false positives [4].

Contrary to the assumption in many existing feature selection methods that SNPs associated with a phenotype are shared across diverse populations, numerous studies highlight population-specific genetic associations with certain diseases [5]. Notably, diseases can manifest distinct prevalence patterns across populations, leading to variations in risk variants from one genetic ancestry to another [6]. For instance, multiple studies underscore that Africans and Europeans exhibit dissimilar genes associated with the lactase-persistence phenotype [7], emphasizing the population-specific nature of genetic influences. Moreover, recent research has revealed significant differences in genetic risk factors of type 2 diabetes among East Asian and European individuals, highlighting the importance of considering population-specific genetic architectures in disease studies [8].

In previous work, we have introduced the Multitask Group Lasso (MuGLasso) framework, designating groups as blocks of SNPs in strong Linkage Disequilibrium (LD) and tasks as sub-populations [9]. We demonstrated its effectiveness in stably identifying SNPs associated with breast cancer.

Despite its effectiveness, the original MuGLasso design required additional post-processing steps to discern task-specific LD-groups [9], typically by filtering out LD-groups with near-zero coefficients in each task after model training. This limitation prompted the introduction of a second regularization term to enhance population-specific sparsity.

Hence this paper introduces the Sparse Multitask Group Lasso (SMuGLasso), an extension of MuGLasso aimed at refining the population-specific selection of LD-groups. By combining the $\ell_{1,2}$-norm penalty of MuGLasso with an additional $\ell_1$-norm at the LD-group level, SMuGLasso seeks to improve the precision of LD-groups selection.

We evaluate the performance of SMuGLasso against MuGLasso using simulated data and the DRIVE breast cancer dataset. In addition to these qualitative phenotypes, we assess both

MuGLasso's and SMuGLasso's effectiveness on a quantitative *Arabidopsis thaliana* phenotype, further validating our approaches on non-human data with a large number of subpopulations.

Finally, we use enrichment analyzes and protein-protein interaction networks to analyze SMuGLasso's findings on the DRIVE breast cancer data sets, shedding light on the molecular mechanisms underlying breast cancer tumor growth.

Finally, we compare the stability of SMuGLasso, MuGLasso, and other existing methods in identifying LD-groups and SNPs associated with a phenotype, aiming to provide a comprehensive understanding of the proposed framework's advantages in addressing the challenges inherent to GWAS.

## Materials and methods

### Ethics statement

The study used data from the DRIVE Breast Cancer OncoArray Genotypes dataset (dbGaP Study Accession phs001265.v1.p1), obtained from NIH after ethical review of project #17707, titled "Network-guided multi-locus biomarker discovery", and used under approval of this request (#67806-4).

### General framework

We introduce SMuGLasso, a four-step framework designed to enhance the precision of population-specific causal variants selection. The steps are similar to those of MuGLasso and are outlined as follows:

1. **Populations assignment**: Each sample is assigned to a genetic population using PCA and k-means clustering. This results in the assignment of each population to an input task within the multitask framework, facilitating a tailored analysis for distinct subpopulations.

2. **LD-Groups formation**: LD-groups consisting of strongly correlated SNPs are formed using adjacency-constrained hierarchical clustering through the `adjclust` [10] package to alleviate the curse of dimensionality by conducting feature selection at the group level. More specifically, we performed this clustering on each chromosome and subpopulation before merging the common boundaries across populations to construct a common set of shared LD groups (see S1 Appendix). Note that LD groups are non-overlapping and exhaustive, meaning each SNP is assigned to exactly one LD group and does not appear in multiple groups.

3. **Model fitting with dual penalty**: The model is fitted with a regularization comprising two penalty terms. Firstly, the MuGLasso penalty involves an $\ell_{1,2}$-norm, fostering sparsity at the LD-group level across all tasks and populations. However, this penalty does not promote sparsity at the task/population level, and does not allow to identify population-specific LD-groups. For this reason, we add in SMuGLasso a second $\ell_1$-norm penalty to enforce sparsity, specifically at the LD-group level for individual populations. To address computational complexity, the optimization problem is solved using coordinate descent with gap safe screening rules [11].

4. **Stability selection**: To improve the robustness of the algorithm, we incorporate a stability selection procedure [12] to ensure a more stable genetic variants selection, contributing to the overall resilience of SMuGLasso.

Unlike MuGLasso, the proposed setting eliminates the need for additional post-processing steps to obtain population-specific LD-groups. In MuGLasso, such population-specific groups

are identified through a post-processing step, which removes LD-groups with near-zero coefficients for each task. SMuGLasso stands out by offering a more precise and refined approach to the selection of population-specific causal SNPs, thereby streamlining the process and enhancing the accuracy of the analysis.

As the population assignment, LD-groups formation, and stability selection procedures are similar to those presented in MuGLasso, we refer the reader to [9] for details and proceed with a detailed discussion of the SMuGLasso model fitting itself. The details of the model and its implementation are provided in S1 Appendix.

## Notations

Given a set of $p$ SNPs measured on $n$ samples, we split the $n$ samples in $T$ subpopulations/tasks, each of size $n_t$ for $t = 1, \dots, T$, and the $p$ SNPs in $G$ LD-groups, each of size $p_g$ for $g = 1, \dots, G$. For each population $t$, we denote by $\boldsymbol{x}_m^{(t)}$ the $p$-dimensional vector of SNPs of the $m$-th sample in the population ($m = 1, \dots, n_t$), and by $y_m^{(t)}$ its phenotype. In practice, we consider SNPs encoded as 0, 1 or 2 depending on the number of reference alleles, but the framework applies to any one-dimensional encoding.

## MuGLasso and its formulation

In what follows, we recall the formulation of MuGLasso as presented in [9]. MuGLasso leverages a penalized regression framework to model the relationship between SNPs and phenotypes. The formulation seeks to achieve sparsity at the LD-group level and smoothness of regression coefficients within and across tasks. We formulate MuGLasso optimization problem as follows:

$$\min_{B \in \mathbb{R}^{p \times T}} \frac{1}{n} \sum_{t=1}^{T} \left( \sum_{m=1}^{n_t} \mathcal{L} \left( y_m^{(t)}, \sum_{j=1}^{p} \beta_j^{(t)} x_{mj}^{(t)} \right) \right) + \lambda \sum_{g=1}^{G} \sqrt{p_g} \left\| B^{(g)} \right\|_F, \tag{1}$$

where $\beta^{(t)} \in \mathbb{R}^p$ represents the regression coefficients specific to task $t$, denoted as $\beta^{(t)} = \left( B_{1t}, \dots, B_{pt} \right)$, so that $B_{jt} = \beta_j^{(t)}$ represents the effect of SNP $j$ in task $t$. The loss function $\mathcal{L}$ takes the form of quadratic loss for quantitative phenotypes ($y \in \mathbb{R}$) and logistic loss for qualitative phenotypes ($y \in 0, 1$). Here, the outer summation is over tasks ($t = 1, \dots, T$), and the inner summation is over the individuals ($m = 1, \dots, n_t$) within each task, which makes explicit the two-level structure of the data: individuals nested within populations. The Frobenius norm $\| \cdot \|_F$ is used to quantify the size of matrices, and $B^{(g)} \in \mathbb{R}^{p_g \times T}$ refers to the submatrix of the full coefficient matrix $B \in \mathbb{R}^{p \times T}$, corresponding to the $p_g$ SNPs in LD group $g$ across all $T$ tasks.

MuGLasso can be reformulated by transforming the original dataset into a new one, represented as a block-diagonal matrix denoted as $(\tilde{X}, \tilde{y})$. This reformulation is introduced to express our problem in a standard single-task regression framework that supports structured penalties such as group-lasso. Unlike typical multitask learning settings, where each task shares the same samples but predicts different phenotypes, our setup involves different input populations predicting a singlesame phenotype. Stacking the design matrices $X^{(t)}$ block-diagonally into $\tilde{X}$ and concatenating the phenotype vectors into $\tilde{y}$ enables compatibility with common optimization algorithms and better designs input multiple tasks. Here, $\tilde{X} \in \mathbb{R}^{n \times pT}$ forms a block-diagonal matrix where each of the $T$ diagonal blocks corresponds to the SNP matrix $X^{(t)} \in \mathbb{R}^{n_t \times p}$ for task $t$. Additionally, $\tilde{y}$ is an $n$-dimensional vector obtained by stacking the phenotype vectors for each task. Introducing this transformation, we derive an adjusted

optimization problem. Let $\boldsymbol{b} \in \mathbb{R}^{p^T}$ be the vector of regression coefficients, where $p^T$ represents the total number of features across all tasks. The reformulated optimization problem is then:

$$\min_{b \in \mathbb{R}^{p^T}} \frac{1}{n} \sum_{i=1}^{n} \mathcal{L}\left(\tilde{y}_i, \sum_{k=1}^{p^T} b_k \tilde{x}_{ik}\right) + \lambda \sum_{g=1}^{G} \sqrt{p_g} \left\| \boldsymbol{b}^{(g)} \right\|_2,$$

where $\tilde{y}_i$ is the $i$-th entry of the transformed phenotype vector $\tilde{y}$, and $\tilde{x}_{ik}$ is the $(i,k)$-th entry of the block-diagonal matrix $\tilde{X}$. Additionally, $\boldsymbol{b}^{(g)} \in \mathbb{R}^{p_g T}$ denotes the regression coefficients associated with SNPs of group $g$, and $\sqrt{p_g}$ is the square root of the size of group $g$.

## SMuGLasso

### Problem formulation

To address potential limitations of the MuGLasso framework, we introduce the SMuGLasso method, which incorporates an additional $\ell_1$ penalty to the original formulation. This additional penalty aims to better control the selection of LD groups across different populations. MuGLasso applies the same group-level penalty for all tasks, but it cannot directly exclude a group from one population while keeping it in another—this requires a manual post-processing step to identify near-zero coefficients. In contrast, the SMuGLasso formulation encourages sparsity at the population level: it can automatically select or remove an LD group for a specific population during training. This leads to more accurate identification of population-specific variants and avoids extra steps after fitting the model. The optimization problem of SMuGLasso is written as follows:

$$\min_{B \in \mathbb{R}^{p \times T}} \underbrace{\frac{1}{n} \sum_{\substack{t=1 \\ \text{(tasks)}}}^{T} \left( \sum_{\substack{m=1 \\ \text{(samples in } t)}}^{n_t} \mathcal{L}\left(y_m^{(t)}, \sum_{j=1}^{p} \beta_j^{(t)} x_{mj}^{(t)}\right) \right) + \lambda_1 \sum_{g=1}^{G} \sqrt{p_g} \left\| B^{(g)} \right\|_F}_{\text{MuGLasso}}$$

$$+ \lambda_2 \sum_{g=1}^{G} \sqrt{p_g} \left\| B^{(g)} \right\|_1, \tag{2}$$

The penalization parameters $\lambda_1$ and $\lambda_2$ control the respective strengths of the two regularization terms. Here, $\|B^{(g)}\|_1$ denotes the elementwise $\ell_1$ norm of the matrix $B^{(g)}$, defined as $\sum_{j=1}^{p_g} \sum_{t=1}^{T} |B_{jt}^{(g)}|$. This penalty encourages sparsity within LD-groups at the population level, complementing the group-wise selection enforced by the Frobenius norm. The combination of the group-level Frobenius norm and the elementwise $\ell_1$ norm allows the model to identify LD-groups that are relevant across tasks while discarding irrelevant coefficients for specific populations. This improves interpretability by distinguishing shared from population-specific signals, and helps reduce false positives by avoiding the selection of entire groups when only a few elements are relevant.

### Optimization

The formulation of SMuGLasso can be transformed exactly as that of MuGLasso shown above. The reformulated model is then:

$$\min_{b \in \mathbb{R}^{p^T}} \frac{1}{n} \sum_{i=1}^{n} \mathcal{L}\left(\tilde{y}_i, \sum_{k=1}^{p^T} b_k \tilde{x}_{ik}\right) + \lambda_1 \sum_{g=1}^{G} \sqrt{p_g} \left\| \boldsymbol{b}^{(g)} \right\|_2 + \lambda_2 \sum_{g=1}^{G} \sqrt{p_g} \left\| \boldsymbol{b}^{(g)} \right\|_1, \tag{3}$$

where $\boldsymbol{b}^{(g)} \in \mathbb{R}^{p_g T}$ is the vector of regression coefficients corresponding to all SNPs of group $g$ for all tasks.

## Gap safe screening rules

Gap safe screening [11] is a method designed to enhance the efficiency of solving regularization problems in statistical learning and high-dimensional data analysis. Hence, gap-safe employs a set of rules to identify and eliminate irrelevant features from the optimization problem, significantly reducing computational complexity. These rules use duality gaps to rigorously guarantee that the discarded features have zero coefficients in the optimal solution, ensuring the accuracy of the model while improving computational speed. This approach is particularly useful in cases with large datasets and numerous features, making it a valuable tool in GWAS data analysis. We have detailed the fundamentals of these rules in MuGLasso paper [9]. Code is available in https://github.com/asmanouira/SMuGLasso.

## Related work

Our method is related to the group Lasso and multitask Lasso, which both rely on an $\ell_{1,2}$-norm penalty [13,14]. Building on that, several studies have been proposed related to multitask variants composed of either two or three regularization terms [15–18]. Notably, these models exhibit limitations in scalability when confronted with high-dimensional data, rendering them inapplicable to our specific context.

To effectively select the additional population-specific regularization term for SMuGLasso, we conducted a comprehensive investigation into the applicability of existing methods. Notably, we found that the proposed sparsity-enforcing penalties were not suited to our specific problem. Our objective is to implement a regularization term that enforces sparsity for specific populations at the level of LD-groups.

We specifically examine the method proposed by (Li L, et al.) [18], which suggests implementing three regularization-based multitask models. Their optimization problem is reformulated as follows:

$$\min_{\beta \in \mathbb{R}^{p \times k}} \frac{1}{2} \sum_{t=1}^{T} \sum_{m=1}^{n_t} \left\| y_m^{(t)} - \sum_{j=1}^{p} \beta_j^{(t)} x_{mj}^{(t)} \right\|_2^2 + \lambda_1 \underbrace{\sum_{j=1}^{p} \left\| \beta_j \right\|_2}_{\mathcal{R}_1(\beta)} + \lambda_2 \underbrace{\sum_{t=1}^{T} \sum_{g=1}^{G} \sqrt{p_g} \left\| \beta_g^{(t)} \right\|_2}_{\mathcal{R}_2(\beta)}$$
$$+ \lambda_3 \underbrace{\sum_{t=1}^{T} \left\| \beta^{(t)} \right\|_1}_{\mathcal{R}_3(\beta)}. \tag{4}$$

Here, the authors aim to enforce population-specific group sparsity using the term $\mathcal{R}_2(\beta)$, aiming to select certain groups only for specific tasks or subpopulations. However, relying only on this regularization term results in the optimization problem being separated across tasks, meaning that the selection is performed independently for each single task. To ensure simultaneous task fitting, the authors introduce the term $\mathcal{R}_1(\beta)$, corresponding to multitask regularization at the single-SNP level across all $T$ tasks. Additionally, they seek to enforce sparsity within groups using an $\ell_1$-norm over all SNPs, represented by a third regularization term (defined by $\mathcal{R}_3(\beta)$), corresponding to the second regularization term of the well-known sparse group Lasso.

In our study, we aim to enhance the selection for population-specific LD-groups. Hence, as mentioned above integrating $\mathcal{R}_2(\beta)$ into SMuGLasso would not maintain multitasking across the tasks $T$, while implementing $\mathcal{R}_3(\beta)$ alongside SMuGLasso would impede the interpretability of the selected features. Selecting SNPs within groups for specific populations complicated determining the number of SNPs within an LD-group $g$ that must be set to 0 to consider the group as not selected for a particular task $t$. Additionally, the inclusion of two penalties in SMuGLasso substantially increased computational demands at a GWAS scale.

Another approach has also been proposed presenting a sparse group multitask feature selection model for GWAS data aimed at leveraging pleiotropy, i.e., SNPs associated with multiple complex diseases [19]. However, it's important to note that their method addresses a different scenario from ours. Specifically, they focus on scenarios where the tasks are output phenotypes rather than input populations samples. In their setting, the goal is to select groups of SNPs targeting the same gene or pathway. Additionally, they combine multiple GWAS datasets and retain only the SNPs shared between them, resulting in a substantial reduction in the number of SNPs (down to 3,766 SNPs) and thereby reducing computational complexity significantly.

## Experiments

### Data

**Simulated data.** Using GWAsimulator [20], we simulate GWAS data following LD patterns of two populations (CEU: Utah residents with Northern and Western European ancestry and YRI: Yoruba in Ibadan, Nigeria) from HapMap3 [21]. We generate different numbers of samples through subpopulations to mimic the structure of real data, where samples through subpopulations are not necessarily equally distributed. We also produce the population stratification confounder by varying the case:control ratio within each subpopulation (CEU 1 300:1 700 and YRI 400:600).

We predefine a total of 200 causal SNPs (non-null hypotheses) as shown in Table 1, in which 50 SNPs (respectively 50 SNPs) are specific to the CEU (respectively YRI) and 100 shared between both populations. All other SNPs were considered non-causal (null hypotheses). We locate the predefined disease loci (and their corresponding LD-groups) randomly and without loss of generality through chromosomes 12, 19, 21, and 22, as shown in Table 2. In total, the data is composed of 4 000 samples and 50 000 SNPs. For CEU, there are 1,407 LD-groups, each containing an average of 35 SNPs. For YRI, there are 995 LD-groups, each containing an average of 50 SNPs. To evaluate method performance, we considered a SNP (or LD-group) as a true positive if it overlapped with any of the predefined causal SNPs (or LD-groups containing them). Conversely, selected SNPs or LD-groups not overlapping with any causal loci were counted as false positives. Note that we used a single simulation replicate due to the high computational cost of running all comparison methods with stability selection on large-scale data (4,000 samples and 50,000 SNPs).

Table 1. For simulated data, number of predefined causal SNPs.

| Populations | Number of SNPs |
| --- | --- |
| Specific-CEU | 50 |
| Specific-YRI | 50 |
| Shared (CEU+YRI) | 100 |
| Total | 200 |

**Table 2. For simulated data, location of predefined disease loci represented by start/end positions and its corresponding LD-groups number in each subpopulation through chromosomes: 12, 19, 21 and 22.**

| Chromosome | Subpopulations | |
|---|---|---|
| | CEU | YRI |
| | loci (# LD-groups) | loci (# LD-groups) |
| 12 | 4 000 - 4 050 (3) | 4 000 - 4 050 (1) |
| 19 | 1 000 - 1 050 (2) | 1 000 - 1 050 (2) |
| 21 | ∅ | 10 000 - 10 050 (1) |
| 22 | 1 000 - 1 050 (3) | ∅ |

**DRIVE breast cancer OncoArray.**   The DRIVE OncoArray dataset contains 28 281 individuals that were genotyped for 582 620 SNPs. 13 846 samples are cases and 14 435 are controls. The dataset contains data for the following countries: USA, Uganda, Nigeria, Cameroon, Australia and Denmark. Additional information about data access and ethical approval is presented in S2 Appendix.

*Arabidopsis thaliana.*   Our *Arabidopsis thaliana* dataset comes from the 1001 Genomes Project [22] (Build TAIR10). We study the DTF3 phenotype, which is the time until the first open flower, in days. The dataset obtained from easyGWAS [23] contains 923 samples and 6 973 565 SNPs divided into 5 chromosomes. This dataset contains plant samples coming from 44 countries.

## Preprocessing

**Quality control and imputation.**   For the simulated dataset and DRIVE breast cancer, we exclude SNPs with a minor allele frequency below 5%, a p-value for Hardy-Weinberg Equilibrium in controls below $10^{-4}$, or a genotyping rate missing more than 10%. We also remove duplicate SNPs, as well as samples with over 10% missing SNPs. We impute missing genotypes in DRIVE using IMPUTE2 [24].

For *Arabidopsis thaliana*, we perform the quality control steps recommended by [23]. We use a Box-Cox transformation [25] of the phenotype to improve the measurements normality. We remove SNPs with a minor allele frequency lower than 5%.

**LD pruning.**   We perform LD pruning using PLINK [26] with an LD cutoff of $r^2 > 0.85$ and a sliding window of 50Mb for the simulated data and DRIVE. For *Arabidopsis thaliana*, we use an LD cutoff of $r^2 > 0.75$ and a window size of 50Mb. After preprocessing steps, 50 000 SNPs remain in the simulated data, 312 237 SNPs in DRIVE and 564 291 SNPs in the *Arabidopsis thaliana* data.

**Population structure.**   We use PLINK [26] to compute the principal components of the genotype matrix. In the simulated dataset, we find two populations, corresponding to the CEU and YRI populations (see S1 Fig). In DRIVE, we identify two populations (see S2 Fig) that we call in this paper POP1 (samples from the USA, Australia and Denmark) and POP2 (samples from the USA, Cameroon, Nigeria and Uganda).

In the *Arabidopsis thaliana* dataset, among 44 countries, we retrieve 5 populations using k-means clustering of the top 4 principal components (see S3 Fig, S4 Fig and S5 Fig). These 5 populations are detailed in S1 Table.

**LD-groups choice.**   For simulated and DRIVE data, we determine the LD-groups for each subpopulation and each chromosome using `adjclust` [10]. However, for *Arabidopsis thaliana*, `adjclust` did not scale computationally to the huge number of SNPs in the five chromosomes. Thus, we first split each chromosome into independent LD-blocks using

snpldsplit [27] function from bigsnpr R package [28]. We then form the LD-groups by applying adjclust on the obtained chunks of independent LD-blocks.

For all three datasets, we then combine these LD-groups across populations by merging their boundary coordinates to obtain shared LD-groups.

Table 3 shows the number of LD-groups obtained for each subpopulation and the final number of shared groups.

## Comparison partners

As a baseline, we use PLINK to conduct association studies between each SNP individually and the phenotype, employing either the top PCs as covariates (**Adjusted GWAS**) or treating each population separately (**Stratified GWAS**). We also add **FastLMM** [29] as a representative baseline of linear-mixed model approaches. FastLMM is designed to adjust globally for population stratification but cannot identify population-specific SNPs; note also that there is no point in running it separately on each subpopulation.

Additionally, we derive a PCA-adjusted phenotype by regressing the top PCs against the phenotype to compute the residuals. To explore the impact of grouping correlated SNPs, segregating populations into tasks, and using an additional penalty to automatically select population-specific SNPs, we compare SMuGLasso with MuGLasso and various other methods. These include a single-task Lasso without groups applied to each population separately (**Stratified Lasso**) or the adjusted phenotype (**Adjusted Lasso**), as well as a single-task group Lasso **Stratified group Lasso** and **Adjusted group Lasso** applied similarly with an adjusted phenotype. (Details in Table 4).

**Table 3**. **Number of LD groups for each subpopulation of the studied datasets (simulated, DRIVE and** *Arabidopsis thaliana*), **and after combination across subpopulations.**

| Data | Subpopulations | # LD-groups | # shared LD-groups |
|---|---|---|---|
| Simulated data | CEU | 1 407 | 1 566 |
| | YRI | 995 | |
| DRIVE real data | POP1 | 8 152 | 17 782 |
| | POP2 | 5 032 | |
| *A. thaliana* data | POP1 | 1 846 | 7 080 |
| | POP2 | 1 950 | |
| | POP3 | 2 002 | |
| | POP4 | 1 728 | |
| | POP5 | 1 834 | |

**Table 4**. **Summary of baseline methods compared to SMuGLasso.** *Population Handling*: **Separated** means each population is analyzed independently; **Handled together** indicates populations are pooled into a single dataset; **Modeled jointly** refers to multi-task learning where populations are treated as related but distinct tasks.

| Method | Population Handling | LD Grouping | Adj. Pheno |
|---|---|---|---|
| Adjusted GWAS | Handled together | No | No |
| FastLMM | Handled together | No | No |
| Stratified Lasso | Separated | No | No |
| Adjusted Lasso | Handled together | No | Yes |
| Stratified Group Lasso | Separated | Yes | No |
| Adjusted Group Lasso | Handled together | Yes | Yes |
| MuGLasso | Modeled jointly | Yes | No |
| SMuGLasso | Modeled jointly | Yes | No |

Note that for **Adjusted Lasso** and **Adjusted Group Lasso**, when handling qualitative phenotypes (such as case-control in DRIVE), we employ logistic regression on the top principal components (PCs) for adjustment. The resulting residuals are subtracted from the actual phenotype values (1 for cases or 0 for controls) to generate a newly adjusted phenotype [30]. This adjustment method, while not widely used, was chosen because it effectively reduced population stratification effects in our experiments, achieving an inflation factor close to 1.

Similarly, for quantitative phenotypes (e.g., DTF3 in *Arabidopsis thaliana*), we apply the same procedure but with linear regression. Traditionally, PCA-based methods or linear mixed models are commonly used for population stratification adjustment in GWAS. With such approaches, it is always possible to add top PCs as additional features, but there is no guarantee that the method will select, and therefore use, them. For instance, FastLMM [29] is recommended for *Arabidopsis thaliana*. However, integrating linear mixed models for feature selection poses challenges in machine learning applications.

In practice, we use bigLasso [31] for the lassos and gap safe screening rules [11] for the group Lasso to optimize computational efficiency. Across all methods, we determine the regularization hyperparameter(s) through cross-validation. More specifically, we use f1-score as a criterion for binary phenotypes and RMSE for quantitative phenotypes. We select the hyperparameters $\lambda_1$ and $\lambda_2$ using 3-fold cross-validation on simulated data generated with GWASimulator, which uses real HapMap3 CEU and YRI genotypes, closely matching real datasets like DRIVE. Due to computational constraints, we reused these tuned values for the DRIVE dataset. For *Arabidopsis thaliana*, we performed 3-fold cross-validation directly on the dataset (without stability selection), and found that the selected hyperparameters were similar to those from simulation-based tuning. To ensure a fair comparison, 3-fold cross-validation was consistently used across all datasets and methods. For models that jointly handle all populations (e.g., SMuGLasso, MuGLasso, Adjusted Lasso/Group Lasso), folds were stratified by both phenotype and population label to preserve balance. For methods applied independently per population (Stratified Lasso/Group Lasso), folds were stratified only by phenotype within each subpopulation.

To assess methodological performance, we analyze runtime, the ability to identify true causal SNPs (in simulated data), and the stability of feature selection. To quantify the stability of the feature selection procedure with respect to perturbations of the input, we repeat the feature selection process on 10 subsamples of the data and report the average Pearson's correlation among all pairs of indicator vectors representing the selected features for each subsample (see S1 Appendix). Our choice of Pearson's correlation is based on the benchmark of Nogueira et al. [32], which shows it is superior to alternatives such as the Jaccard index as it adjusts for chance agreement and allows for selection sizes that vary across repetitions.

## Biological interpretation

**Functional mapping and annotations analysis.** We use FUMA [33] to map functionally annotated SNPs to genes according to the physical position in the genome and eQTL mapping. The tool uses information from multiple biological data to perform these mapping analyses. We used Ensembl version 110 as the reference, and the 1000 Genome Project/Phase3 as the reference panel. A physical mapping window of 10 kb was employed to map variants to nearby genes, a commonly used range to capture potential regulatory effects of nearby SNPs while limiting false positives. We ensured that FUMA would interpret all the SNPs we had selected by assigning them an arbitrary and fake p-value below the significance threshold of $5.10^{-8}$ it uses for selecting SNPs of interest. The eQTL mapping was performed using GTEx data version 6, including all available tissue types.

For *Arabidopsis thaliana* dataset, we map SNPs identified by Adjusted GWAS, MuGLasso, and SMuGLasso to genes using TAIR10, which provides genomic location data of *Arabidopsis thaliana* genes in GFF3 format.

**Gene set enrichment analysis.** We use Metascape [34] to perform gene set enrichment analysis to understand the functional relevance of the obtained gene lists that Adjusted GWAS, MuGLasso and SMuGLasso have discovered. This tool performs pathway and process enrichment analysis using multiple pathway data bases: KEGG Pathway, Reactome Pathway, WikiPathways, PID, BioCarta, Panther Pathway, SMPDB, GO Biological Processes, CORUM, TargetScan Pathway, TF targets and PharmGKB. Metascape also performs gene set enrichment analysis against cell type signatures, the gene-disease database DisGeNET, the pattern gene database PaGenBase, and transcription factor targets. We selected Metascape over other tools because it integrates a wide range of annotation resources in a single platform, offers high-quality visualizations, and supports both human and non-human species, including *Arabidopsis thaliana*.

Metascape collects terms with an enrichment p-value <0.01, a minimum number of occurrences of 3, and an enrichment factor >1.5 and groups them into clusters based on membership similarities. The term with the smallest p-value within a cluster then represents the cluster.

**Protein-protein interaction network analysis.** This analysis consists of an additional layer of biological validation and helps assess whether the selected genes are not only statistically significant, but also biologically connected through shared functional pathways. We further use Metascape [34] to construct a protein-protein interaction network between enriched genes. Metascape uses multiple sources to this end, including experimental and predicted interactions, and identifies densely connected network components using MCODE [35]. We finally visualize the obtained network using Cytoscape [36] to enable the discovery of functionally related gene groups within breast cancer disease on the DRIVE dataset and on within DTF3 on the *Arabidopsis thaliana* dataset.

## Results

### SMuGLasso and MuGLasso rely on both LD-groups and the multitask approach to recover disease SNPs in simulated data

On simulated data, we observe that SMuGLasso and MuGLasso outperform the other methods at recovering the predefined disease loci (See Fig 1). In addition, we confirm that performing feature selection at the level of LD-groups provides better performance compared to the conventional single-SNP selection. This confirms that grouping SNPs helps to alleviate the curse of dimensionality and improve the identification of causal variants.

Table 7 gives, for both SMuGLasso and MuGLasso, the number of selected LD-groups and SNPs across and per subpopulation for each dataset. Compared to MuGLasso, we notice that SMuGLasso ensures more sparsity for shared selection across all subpopulations thanks to its additional $\ell_1$-norm penalty.

SMuGLasso provides a more precise selection for population-specific level. Indeed, SMuGLasso successfully recovers causal LD-groups/SNPs that MuGLasso missed in simulated data (see Fig 2).

We note that SMuGLasso is more intensive computationally compared to MuGLasso and any other tested method (see Fig 3). This computational cost is caused by the additional population-specific regularization term. However, the implementation is efficient enough to

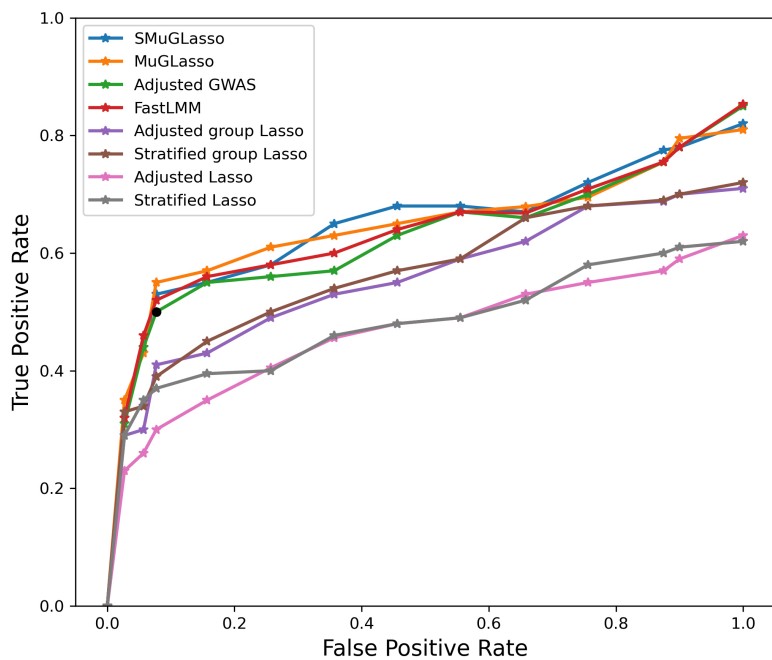

**Fig 1. On simulated data, ability of different methods to retrieve causal disease SNPs as a ROC plot.**

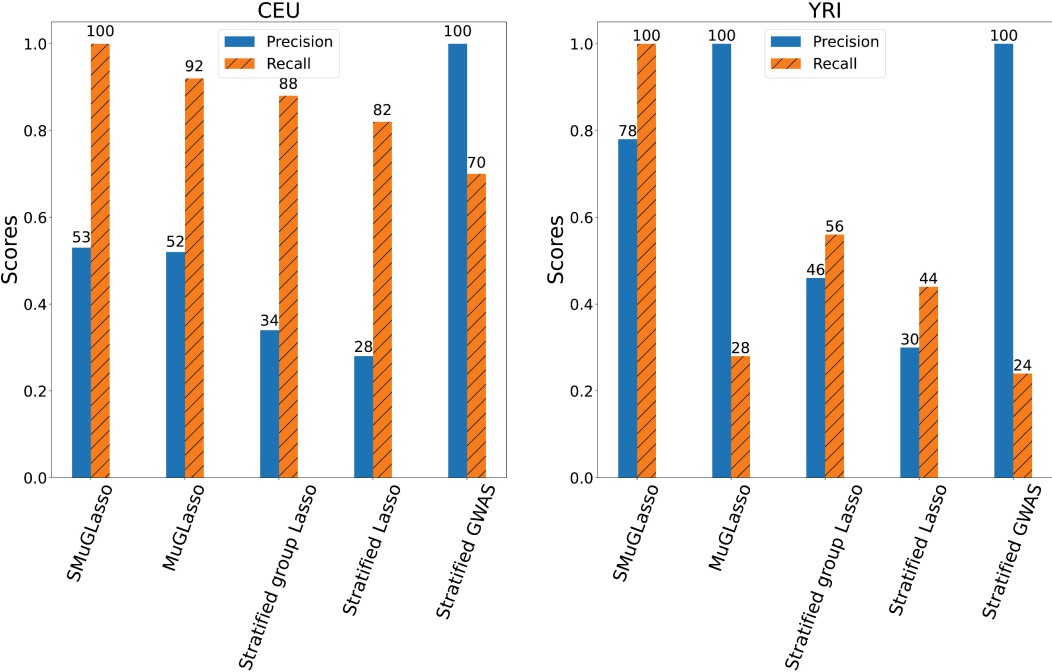

**Fig 2. For simulated data, precision and recall of SMuGLasso, MuGLasso and the stratified approaches on the populations-specific SNPs.**

scale to high-dimensional GWAS data thanks to gap-safe screening rules. Here, the reported runtimes for both SMuGLasso and MuGLasso include the stability selection procedure.

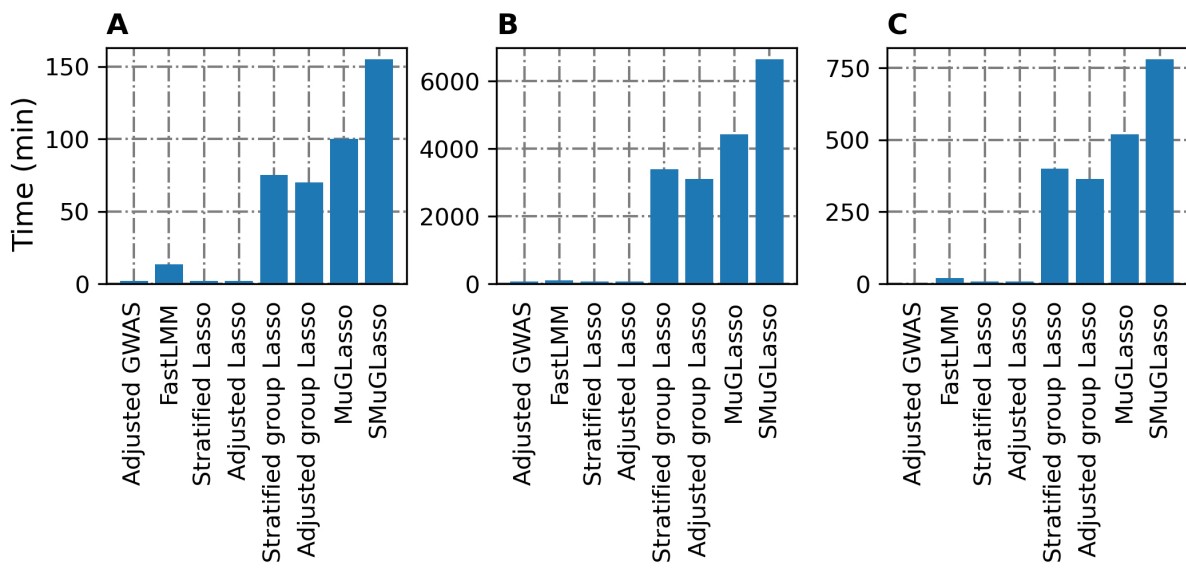

**Fig 3. Runtimes of Lasso approaches for simulated, DRIVE and *Arabidopsis thaliana* datasets.**

## MuGLasso and SMuGLasso point to genes of interest not identified by classical GWAS in real data

**Genes identified by physical mapping annotation of SNPs selected in the DRIVE data.** The breast cancer risk genes identified by physical mapping of the SNPs selected by adjusted GWAS, SMuGLasso and MuGLasso on DRIVE are detailed in S2 Table.

SMuGLasso and MuGLasso both recover the 9 risk genes identified by classical GWAS. SMuGLasso identifies 27 more risk genes. Of those, 17 have been previously identified in a meta-GWAS analysis containing the DRIVE data (see S3 Table), and another 8 have been found to be associated with breast cancer risk in other studies, leaving 2 genes with no previous evidence supporting their relation to the disease (see S4 Table).

MuGLasso selects the same genes as SMuGLasso, and an additional 5 genes. We have found in the literature evidence of the association with breast cancer of only 2 of those, leaving another 3 genes with no previous supporting evidence.

To summarize these findings, the distribution of evidence types for genes identified by each method is illustrated in Fig 4.

**Genes identified by physical mapping annotation of SNPs selected in the *Arabidopsis thaliana* data.** The genes associated with the *Arabidopsis thaliana* DTF3 phenotype according to adjusted GWAS, SMuGLasso and MuGLasso are presented in S5 Table. Again, SMuGLasso and MuGLasso both recover all the risk genes (7 in total) identified by classical GWAS. SMuGLasso identifies an additional 41 genes, including 8 population-specific findings, and MuGLasso identifies 7 more genes on top of those selected by SMuGLasso. Only 4 of the 55 genes selected by MuGLasso are population-specific.

**Genes identified by eQTL mapping annotation of SNPs selected in the DRIVE data.** In addition to physical mapping, our use of eQTL functional annotations aims to uncover supplementary information about the genetic basis of breast cancer disease in DRIVE. S6 Table presents the genes obtained by both physical and eQTL mapping of the loci identified by the adjusted GWAS, SMuGLasso and MuGLasso. Using eQTL mapping adds 25 genes to the 9 identified by physical mapping of the adjusted GWAS results. 2 of those (PTHLH and

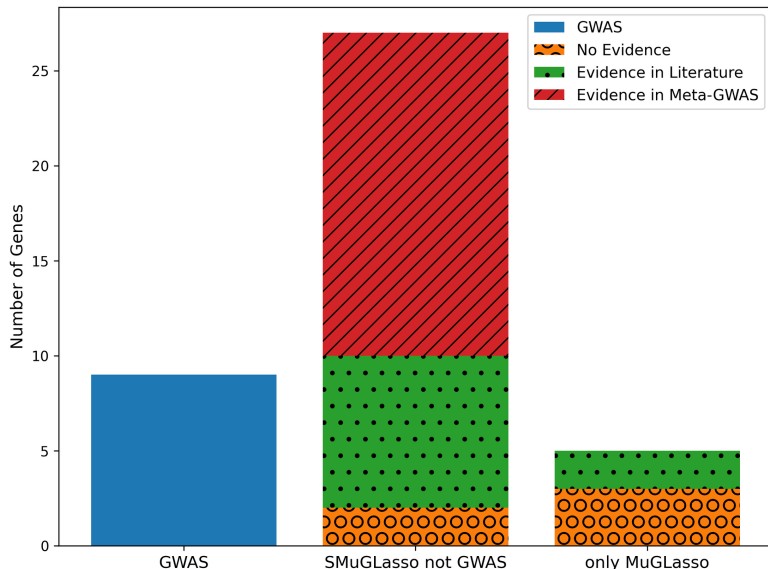

**Fig 4. On DRIVE and through physical mapping annotation: stacked bar plot presenting the number of genes identified by different methods on the x-axis (GWAS genes, shared SMuGLasso and MuGLasso genes not identified by GWAS and only MuGLasso genes not identified by SMuGLAsso and GWAS), and categorized by evidence type (No evidence, evidence in literature and evidence in Meta-GWAS).** Note that MuGLasso finds all genes selected by SMuGLasso, and SMuGLasso finds all genes selected by a classical GWAS.

TNRC6B) had been identified through the physical mapping of loci selected by SMuGLasso (and hence MuGLasso) but not the classical GWAS.

Using eQTL mapping also adds 30 genes to the list of those selected by SMuGLasso but not the classical GWAS. Of these, 26 are confirmed by the literature, as presented in S7 Table). Finally, the loci selected only by MuGLasso point to an additional 12 genes through eQTL mapping, only 2 of which are linked to breast cancer in the literature. To provide further clarification, the distribution of evidence types for these gene discoveries is illustrated on Fig 5.

## SMuGLasso and MuGLasso outperform the other methods in terms of stability

Table 5, Table 6 and S8 Table show the performance of the tested methods concerning the stability, measured by the stability index alongside the number of selected LD-groups and SNPs, along with their selection level (LD-groups or Single-SNP) respectively for simulated, DRIVE and *Arabidopsis thaliana* datasets. We use 100 subsamples to perform stability selection [12]. Indeed, the obtained metrics highlight that stability selection increases the robustness of SMuGLasso, MuGLasso and Adjusted group Lasso for the three datasets.

For the simulated data (Table 5), SMuGLasso and MuGLasso exhibit noteworthy stability, surpassing other methods. SMuGLasso selects 8 LD-groups and 290 SNPs, demonstrating a stability index of 0.5811. Similarly, MuGLasso selects 10 LD-groups and 363 SNPs with a higher stability index of 0.7015. Even without stability selection, both methods marginally increase the number of selected LD-groups and SNPs compared to classical approaches while

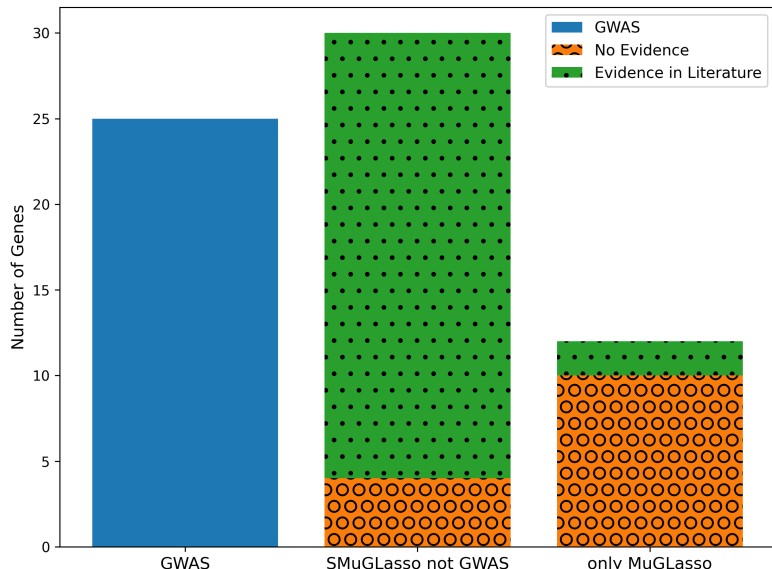

**Fig 5. On DRIVE and through eQTL mapping annotation: stacked bar plot presenting the number of genes identified by different methods in x-axis (GWAS genes, shared SMuGLasso and MuGLasso genes not identified by GWAS and only MuGLasso genes not identified by SMuGLasso and GWAS) and categorized by evidence type (No evidence or evidence in the literature).**

**Table 5. Stability index and number of selected features for different methods, on simulated data.**

| Methods | # selected LD-groups | # selected SNPs | Stability index | Selection level |
|---|---|---|---|---|
| SMuGLasso | 8 | 290 | 0.5811 | LD-groups |
| SMuGLasso without stability selection | 9 | 328 | 0.5045 | LD-groups |
| MuGLasso | 10 | 363 | 0.7015 | LD-groups |
| MuGLasso without stability selection | 11 | 402 | 0.6124 | LD-groups |
| Adjusted group Lasso + stability selection | 11 | 374 | 0.5929 | LD-groups |
| Adjusted group Lasso | 12 | 392 | 0.5340 | LD-groups |
| Stratified group Lasso | 13 | 452 | 0.4491 | LD-groups |
| Adjusted Lasso | 12 | 422 | 0.4053 | Single-SNP |
| Stratified Lasso | 13 | 441 | 0.3140 | Single-SNP |
| Adjusted GWAS | 3 | 109 | 0.9834 | Single-SNP |
| FastLMM | 3 | 73 | 0.9714 | Single-SNP |

maintaining relatively high stability indices. Adjusted GWAS and FastLMM, based on single-marker testing, show even greater stability, with indices of 0.8834 (109 SNPs on 3 LD-groups) and 0.8714 (73 SNPs on 3 LD-groups), respectively.

For the DRIVE dataset (Table 6), SMuGLasso and MuGLasso continue to exhibit better stability than their comparison partners. SMuGLasso selects 58 LD-groups and 1,279 SNPs with a stability index of 0.3881, while MuGLasso selects 62 LD-groups and 1,357 SNPs with a stability index of 0.4312. Adjusted GWAS and FastLMM achieve higher stabilities, with indices of 0.7724 (306 SNPs on 16 LD-groups), and 0.7471 (227 SNPs on 13 LD-groups) respectively.

**Table 6. Stability index and number of selected features for different methods, on the DRIVE data set.**

| Methods | # selected LD-groups | # selected SNPs | Stability index | Selection level |
|---|---|---|---|---|
| SMuGLasso | 58 | 1 279 | 0.3881 | LD-groups |
| SMuGLasso without stability selection | 60 | 1 354 | 0.3325 | LD-groups |
| MuGLasso | 62 | 1 357 | 0.4312 | LD-groups |
| MuGLasso without stability selection | 72 | 1 524 | 0.3911 | LD-groups |
| Adjusted group Lasso + stability selection | 59 | 1 293 | 0.3234 | LD-groups |
| Adjusted group Lasso | 68 | 1 466 | 0.2613 | LD-groups |
| Stratified group Lasso | 58 | 1 119 | 0.2498 | LD-groups |
| Adjusted Lasso | 41 | 874 | 0.2068 | Single-SNP |
| Stratified Lasso | 38 | 789 | 0.1581 | Single-SNP |
| Adjusted GWAS | 16 | 306 | 0.7724 | Single-SNP |
| FastLMM | 13 | 227 | 0.7471 | Single-SNP |

For the *Arabidopsis thaliana* dataset (see S8 Table), SMuGLasso and MuGLasso present once again the best stability indices. SMuGLasso selects 80 LD-groups and 6,367 SNPs with a stability index of 0.4315, while MuGLasso selects 104 LD-groups and 8,254 SNPs with a stability index of 0.5733. Thus, SMuGLasso and MuGLasso demonstrate good performance even with datasets containing relatively few samples. However, Adjusted GWAS (0.7129 for 31 SNPs on 7 LD-groups) and FastLMM (0.8106 for 12 SNPs on 5 LD-groups) again yield the best stability scores.

In summary, traditional GWAS and FastLMM—which rely on single-marker association tests—achieve the highest stability indices across all datasets. This observation reflects the robustness of univariate testing strategies when measuring stability across subsamples [4]. However, it is important to note that these methods tend to detect only a small subset of the causal variants, often restricted to the most strongly associated signals. Moreover, a major limitation of these methods is their inability to identify population-specific associations, as they do not model population structure in a multivariate framework as done in feature selection models. While FastLMM is designed to account for population structure by globally adjusting for diversity or admixture, it does not allow detection of signals specific to individual populations. Similarly, stratifying the analysis in GWAS by population reduces statistical power, often resulting in very few or no detected associations, especially for populations with few samples.

Note that for methods offering selection at the single-SNP level, once a SNP is selected, we consider that the entire LD-group is selected. This approach enables direct comparison across methods with different genomic selection levels (single-SNP level versus LD-groups level). We observe that LD-group-based methods tend to produce more stable results as they reduce the number of candidate options to be selected. Indeed, this makes the selection more consistent across subsamples.

Because MuGLasso does not include an additional $\ell_1$ penalty, it selects more LD-groups than SMuGLasso overall, which affords it higher stability. This is an expected trade-off in sparse modeling, similar to what is observed when comparing Lasso with Elastic Net [37]. MuGLasso remains the model that gives the best stability values on all datasets, followed by SMuGLasso, which outperforms the other applied feature selection methods. It's noteworthy that SMuGLasso produces fewer selected SNPs and LD-groups compared to MuGLasso.

Indeed, enforcing an additional penalty yields a sparser model, at the expense of stability; this behavior is on par with what is usually observed with lasso vs elastic net regularization.

## SMuGLasso and MuGLasso select both population-specific and shared LD-groups on both simulated and real data

SMuGLasso ensures the selection of both shared (across tasks) and task-specific LD-groups. MuGLasso can also provide such a selection at the cost of a post-processing step. Specifically, for each task, LD groups with regression coefficients below a small threshold of $10^{-2}$ are considered inactive and removed. This post-processing was consistently applied to MuGLasso during our comparison experiments to ensure a fair evaluation.

Table 7 presents the number of shared and population-specific LD-groups (along with the corresponding number of SNPs) selected respectively by SMuGLasso and MuGLasso on the simulated, DRIVE, and *Arabidopsis thaliana* data sets. This underscores the ability of both methods to capture task-specific genetic features while also identifying shared patterns across different populations.

For comparison, feature selection in stratified models is conducted separately for each task. Thus, the population-specific LD-groups in stratified models correspond to LD-groups that were only selected in one population. Notably, the adjusted methods for population stratification (Adjusted group Lasso, Adjusted Lasso, and Adjusted GWAS) do not allow the selection of population-specific LD-groups.

These findings have implications for the practical application of the methods. For instance, SMuGLasso has better recall for population-specific SNPs, as illustrated by Fig 2. This figure shows the precision and recall of SMuGLasso, MuGLasso, and the stratified approaches on the population-specific SNPs, highlighting the improved performance of SMuGLasso in reducing the number of falsely selected SNPs, thanks to its additional $\ell_1$-norm regularization.

## Genes selected by SMuGLasso on DRIVE show breast cancer related expression

**Gene set enrichment analysis.** To further explore the usefulness of SMuGLasso, we investigated the genes it selected on the DRIVE dataset by performing gene set enrichment analyses.

**Table 7. Number of selected LD-groups/SNPs, across and per population, for the three data sets, for both SMuGLasso and MuGLasso.**

| Data | Population | # selected LD-groups (and SNPs) | |
|---|---|---|---|
| | | **SMuGLasso** | **MuGLasso** |
| Simulated data | CEU | 2 (104 SNPs) | 2 (88 SNPs) |
| | YRI | 3 (64 SNPs) | 1 (14 SNPs) |
| | shared (CEU and YRI) | 3 (122 SNPs) | 6 (261 SNPs) |
| DRIVE | POP1 | 5 (155 SNPs) | 6 (148 SNPs) |
| | POP2 | 1 (21 SNPs) | 2 (43 SNPs) |
| | shared (POP1 and POP2) | 52 (1 103 SNPs) | 54 (1 166 SNPs) |
| *A. thaliana* | POP1 | 3 (247 SNPs) | 2 (164 SNPs) |
| | POP2 | 5 (381 SNPs) | 4 (303 SNPs) |
| | POP3 | 1 (81 SNPs) | 0 |
| | POP4 | 3 (232 SNPs) | 3 (232 SNPs) |
| | POP5 | 1 (72 SNPs) | 0 |
| | shared (5 populations) | 67 (5 354 SNPs) | 95 (7 555 SNPs) |

S9 Table shows the top 10 pathways and processes enriched in genes selected by SMu-GLasso on DRIVE. These gene sets reveal ontology terms related to biological processes and pathways implicated in breast cancer development and progression, as supported by the literature. For instance, "mammary gland morphogenosis" and "intracellular signaling by second messengers" highlight key pathways involved in breast development and cellular signaling [38,39]. Furthermore, terms such as "ectoderm differentiation" and "endoderm differentiation" point to the importance of cellular differentiation processes in breast tissue homeostasis and tumor formation [40]. Hence, the enrichment of SMuGLasso genes in these pathways underscores their potential roles as regulators in breast cancer biology.

S10 Table, S11 Table, S12 Table, S13 Table present further enrichment analyses against various ontologies. Many of the DisGeNET disease terms that are significantly (corrected p-values < 0.05) enriched in genes selected by SMuGLasso (see S10 Table) correspond to subtypes of breast cancer (estrogen receptor-positive breast cancer, luminal A breast carcinoma, luminal B breast carcinoma, estrogen receptor-negative breast cancer, stage 0 breast carcinoma, mammary neoplasms). Several other enriched disease terms pertain to related diseases (uterine fibroids, squamous cell carcinoma of lung). Finally, the "breast size" trait could indeed be associated with breast cancer risk through breast tissue density, which contributes to an increased risk of developing breast cancer [41].

One cell type signature is significantly (corrected p-value < 0.05) enriched in genes selected by SMuGLasso: fetal thymic epithelial cells (see S11 Table). While the relationship with breast cancer genes is not immediately obvious, this could be related to the role of thymic function in mammary gland development and tumorigenesis [42].

Finally, although not significant after correction for multiple hypothesis testing, transcription factor target enrichment analysis shows enrichment of targets of the Brn-2 transcription factor (see S13 Table). These targets (FGFR2, PTLH, ELL, and ZMIZ1) have already been identified through meta-GWAS of breast cancer (see S3 Table). In addition, this is consistent with findings demonstrating that this transcription factor promotes invasion and metastasis in triple-negative breast cancer cells [43].

**Protein-protein interaction analysis.** We also used Metascape to identify the modules of the protein-protein interaction network formed from known interactions between genes identified through physical mapping of the SNPs selected by the Adjusted GWAS approach, SMuGLasso and MuGLasso, which are shown on Fig 6. The modules obtained when adding genes identified through eQTL mapping can be visualized on S6 Fig. Pathway and process enrichment analysis of the genes in the two modules identified by SMuGLasso (Fig 6C) highlights three significant processes, described in Table 8. These highlight the ability of SMuGLasso to identify more relevant disease genes than a classical GWAS approach. Indeed, as breast cancer often originates from aberrant growth and dysfunction within mammary gland structures, mammary gland morphogenesis processes may serve as crucial drivers or modifiers of tumor initiation and progression [38,44]. Furthermore, phosphorylation is well-known to play a role in regulating cellular processes such as proliferation, migration, and survival, all of which are dysregulated in cancer [45,46] in general and in breast cancer in particular [47].

All in all, gene set enrichment analysis of the genes pinpointed by SMuGLasso highlights the relevance of the disease genes it detects in addition to a classical GWAS, suggesting that this tool can be used to provide valuable insights into the molecular mechanisms underlying phenotypes.

**Quantitative pathway enrichment analysis comparison.** To compare pathway enrichments between Adjusted GWAS, MuGLasso and SMuGLasso, we conduct a quantitative analysis to assess their biological significance. We extract the Z-scores for each common pathway

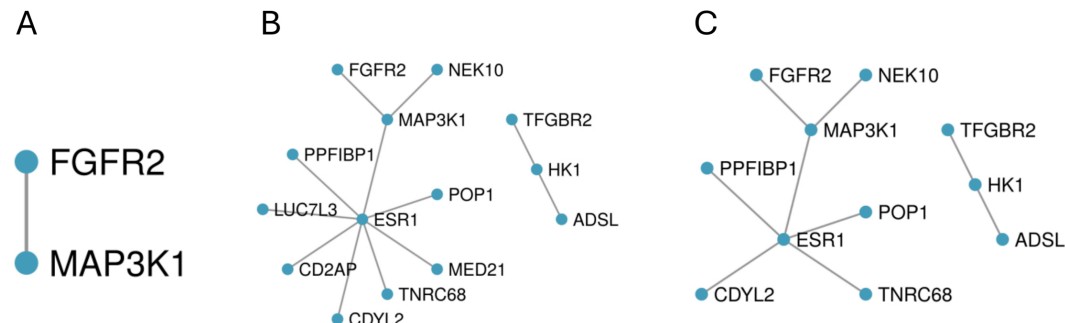

**Fig 6. Modules of the PPI of known interactions between genes identified through physical mapping of the SNPs selected by (A) Adjusted GWAS, (B) MuGLasso and (C) SMuGLasso on DRIVE.**

**Table 8. Pathway and process enrichment analysis of the modules of the PPI of known interactions between genes identified through physical mapping of the SNPs selected by SMuGLasso on DRIVE.**

| GO | Description | Log10(P) | Gene Hits |
|---|---|---|---|
| GO:0060443 | mammary gland morphogenesis | -6.4 | ESR1, FGFR2, TGFBR2 |
| GO:0016310 | phosphorylation | -5.5 | FGFR2, HK1, MAP3K1, TGFBR2, NEK10 |
| GO:0022612 | gland morphogenesis | -5.2 | ESR1, FGFR2, TGFBR2 |

across the three methods (Adjusted GWAS, MuGLasso and SMuGLasso). For this analysis, we consider the top 50 pathways with the highest Z-scores for each method and then focus on the common pathways among these top-ranked sets. For each shared pathway, we compute the Z-score ratio for each pairwise comparison of methods (MuGLasso/GWAS to assess MuGLasso vs. Adjusted GWAS, MuGLasso/Adjusted GWAS to assess Adjusted GWAS vs. SMuGLasso and SMuGLasso/MuGLasso to assess MuGLasso vs. SMuGLasso). Thus, a Z-score ratio higher than 1 indicates that the first method in the pair (e.g., MuGLasso in the MuGLasso/Adjusted GWAS comparison) shows greater enrichment for that pathway than the second method (e.g., Adjusted GWAS). Conversely, a Z-score ratio of less than 1 indicates that the second method in the pair shows greater enrichment for that pathway compared to the first method.

We further assess the significance of differences in pathway enrichments using paired t-tests.

In Fig 7, the box plots illustrate comparisons of pathway enrichments between SMuGLasso, Adjusted GWAS, and MuGLasso. Notably, SMuGLasso shows higher pathway and process enrichment compared to both Adjusted GWAS and MuGLasso, as indicated by the higher Z-score ratios. Furthermore, the performed paired t-tests reveal statistically significant differences between SMuGLasso and MuGLasso, confirming that SMuGLasso identifies pathways with greater biological significance compared to MuGLasso.

Further figures illustrate that SMuGLasso exhibits greater enrichment than MuGLasso across all gene sets (S7 Fig), including DisGeNET gene sets (S8 Fig). We emphasize the DisGeNET gene sets in these analysis, as they are the only ones for which classical GWAS shows enrichment. Compared to GWAS, SMuGLasso demonstrates higher enrichment in 4 out of 6 pathways, equivalent enrichment in 1 pathway, and lower enrichment in 1 other pathway (see S9 Fig), whereas MuGLasso shows better enrichment than GWAS in 3 pathways, equal enrichment in 1 pathway, and lower enrichment in 2 pathways (S10 Fig).

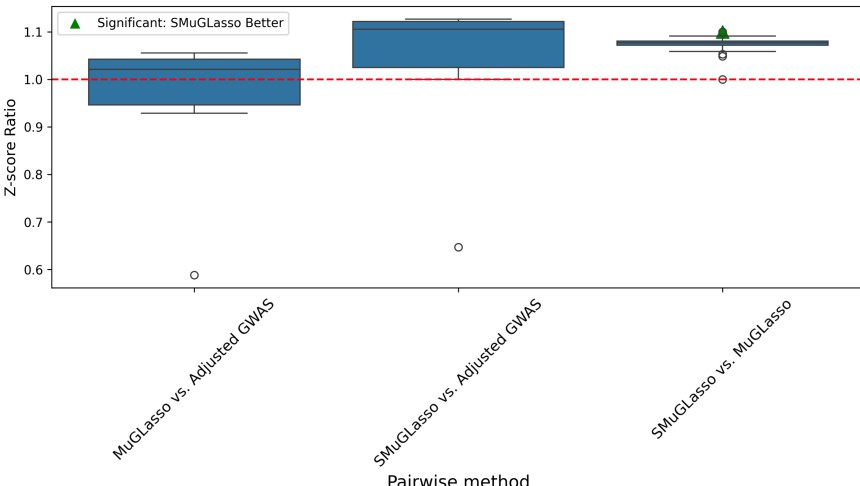

**Fig 7. On DRIVE, box plots representing the distribution of Z-score ratios for gene sets enrichments across three pairwise comparisons: MuGLasso vs. Adjusted GWAS, SMuGLasso vs. Adjusted GWAS, and SMuGLasso vs. MuGLasso.** The Z-score ratio is computed for each shared pathway between the methods. The green triangle indicates significant differences in enrichment (p-value < 0.005) as determined by paired t-tests. The red dashed line at $y = 1$ represents equal enrichment by both methods.

Notably, the top enrichment results determined through pathway analysis are similar regardless of whether or not eQTL gene lists are included.

## Genes selected by SMuGLasso on the *Arabidopsis thaliana* data show flowering time related expression

We present in S5 Fig the list of mapped genes using TAIR10_gff3 mapping of SNPs selected on the *Arabidopsis thaliana* data set by Adjusted GWAS, SMuGLasso and MuGLasso. We conducted pathway enrichment analysis and observed distinct differences in enrichments among the tested methods. SMuGLasso identified 48 genes, among which two pathways are significantly overrepresented: gravitropism and response to carbohydrate. MuGLasso discovered 55 genes, among which only the gravitropism pathway is overrepresented. By contrast, Adjusted GWAS identified 7 genes, for which the enrichment analysis did not yield any pathway.

An advantage of SMuGLasso compared to MuGLasso is that it finds more pathways with fewer gene discoveries, suggesting it is likely more efficient. The pathways identified by SMuGLasso and MuGLasso, namely gravitropism and response to carbohydrate, have potential relevance to the time until the first open flower phenotype. Gravitropism, the orientation or growth of plants in response to gravity, could influence floral development by affecting how plants orient their growth and allocate resources [48]. The response to carbohydrate pathway discovered by SMuGLasso may also be significant, as carbohydrates are crucial for energy storage and signaling, which can impact plant growth and development, including flowering time [49]. These observations suggest that SMuGLasso and MuGLasso methods are more effective in uncovering biologically relevant pathways that could explain variations in the flowering time phenotype compared to Adjusted GWAS, which did not identify any related pathways.

Finally, Fig 8 shows the modules of the protein-protein interaction network of known interactions between the genes identified by SMuGLasso and MuGLasso. The MuGLasso PPI network appears denser and more informative, with additional interactions and nodes such

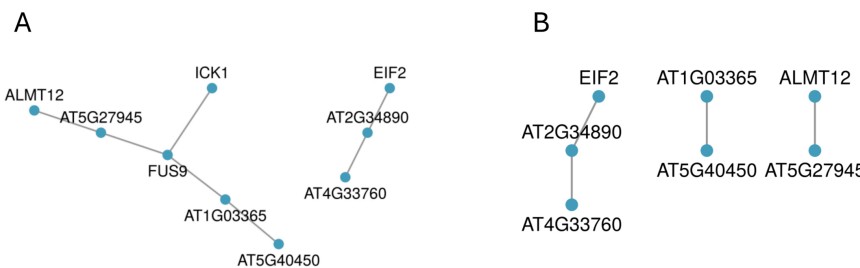

**Fig 8. Modules of the PPI of known interactions between genes identified through physical mapping of the SNPs selected by (A) MuGLasso and (B) SMuGLasso on the *Arabidopsis thaliana* data set.**

as FUS9 and ICK1, compared to the SMuGLasso PPI network, which is sparser and has fewer connections. However, upon further investigation, we did not find any evidence linking these nodes, particularly FUS9 and ICK1, to flowering time regulation in *Arabidopsis thaliana*.

## Discussion

We have presented in this paper SMuGLasso, an extension of MuGLasso for the identification of relevant SNPs from GWAS data across multiple populations. The proposed model is based on a multitask framework in which the tasks are genetic populations and features are clustered in groups. The selection is performed at the scale of LD-groups. The populations are identified using PCA and k-means to assign each sample to a subpopulation. This setting alleviates the curse of dimensionality and addresses population stratification in diverse populations. Compared to MuGLasso, SMuGLasso includes an additional regularization term which enforces task-specific sparsity at the level of LD-groups. Thus, our model provides indeed a more precise recovery of risk regions related to the phenotype at the population-specific level.

Our simulations demonstrate that SMuGLasso outperforms MuGLasso and other methods in accurately identifying population-specific disease loci, while also minimizing potential false discoveries. While MuGLasso shows commendable stability, SMuGLasso closely follows, exhibiting robust stability indexes across various datasets with a reduced number of selected LD-groups/SNPs. The application of stability selection techniques further bolsters SMuGLasso's reliability in terms of stability measurements.

A significant advancement in our study is addressing the computational challenges posed by the additional penalty in MuGLasso, achieved through the implementation of gap-safe screening rules. This ensures efficient processing for both qualitative and quantitative phenotypes.

Lastly, we have detailed the genes identified by both SMuGLasso and MuGLasso in our real data analyses, and we performed pathway analysis with biological interpretation for the entire gene lists. Interestingly, SMuGLasso's findings are more consistent with the literature than those that are specific to MuGLasso. However, we encountered limitations in investigating pathway analysis specific to populations due to the absence of tools that adequately consider population structure. Despite our efforts, we were unable to find evidence in our findings for pathway enrichment specific to particular populations. Looking ahead, our goal is to delve into pathway analysis to unravel the biological mechanisms underpinning the phenotypes of interest in diverse population studies, as revealed by the identified risk genes.

Despite the implementation of gap-safe screening rules, the computational load is still significant, especially when dealing with extremely large datasets. This could limit the applicability in broader GWAS data where computational resources are a constraint. The efficacy of

SMuGLasso heavily relies on the ability of PCA and k-means clustering in identifying subpopulations. Misclassification or suboptimal clustering can potentially impact the final results. One solution is to focus on enhancing the clustering stability through a hierarchical structure. Moreover, given the additional regularization term, there is a risk that the model might become biased towards tasks with more samples, potentially overlooking key insights in the less-represented tasks/populations. Introducing a weighting scheme that balances the influence of each task, particularly giving more weight to those with fewer samples, might help in addressing the imbalance, or integrating additional external datasets to bolster the sample sizes of underrepresented tasks could be beneficial. In addition, investigating different regularization terms for the population-specific LD-groups selection or hybrid approaches could potentially improve the model's performance in identifying disease-relevant loci. Furthermore, there remains an essential avenue for future work in rigorously integrating alternative stability selection methods, which could further improve SMuGLasso's robustness. In conclusion, while SMuGLasso presents a novel framework in the field of GWAS analysis, especially in the precise identification of population-specific risk loci, our ongoing efforts to refine its computational efficiency, enhance clustering accuracy, and balance task representation will be essential in realizing its full potential in unraveling the complex genetic mechanisms of diseases.

## Supporting information

**S1 Appendix. SMuGLasso method details.**
(PDF)

**S2 Appendix. DRIVE.**
(PDF)

**S1 Fig. PCA for simulated dataset.**
(PDF)

**S2 Fig. PCA for DRIVE dataset.**
(PDF)

**S3 Fig. PCA for *Arabidopsis thaliana* dataset.** Projection of the *Arabidopsis thaliana* genotypes on the first two PCA components. Samples originate from 44 countries.
(PDF)

**S4 Fig. PCA for *Arabidopsis thaliana* dataset.** Projection of the *Arabidopsis thaliana*. The identified 5 subpopulations through K-means clustering of the data.
(PDF)

**S5 Fig. Inertia Plot for K-means Clustering of *Arabidopsis thaliana*.**
(PDF)

**S6 Fig. PPI Modules from eQTL and Physical Mapping of GWAS/SMuGLasso-Selected SNPs on DRIVE.** Modules of the PPI of known interactions between genes identified through physical and eQTL mapping of the SNPs selected by Adjusted GWAS, SMuGLasso and MuGLasso on DRIVE.
(TIF)

**S7 Fig. Comparison of All gene set Enrichment (Z-ratios) on DRIVE: SMuGLasso vs. MuGLasso.** On DRIVE, comparison of All gene sets enrichment between SMuGLasso and MuGLasso based on Z-score ratios. Bar heights represent the ratio of Z-scores (SMuGLasso/MuGLasso) for top common gene sets.
(PNG)

**S8 Fig. Comparison of DisGeNET Enrichment (Z-ratios) on DRIVE: SMuGLasso vs. MuGLasso.** On DRIVE, comparison of DisGeNET gene sets enrichment between SMuGLasso and MuGLasso based on Z-score ratios. Bar heights represent the ratio of Z-scores (SMuGLasso/MuGLasso) for top DisGeNET common gene sets.
(PNG)

**S9 Fig. Comparison of DisGeNET Enrichment (Z-ratios) on DRIVE: SMuGLasso vs. Adjusted GWAS.** On DRIVE, comparison of DisGeNET gene sets enrichment between SMuGLasso and Adjusted GWAS based on Z-score ratios. Bar heights represent the ratio of Z-scores (SMuGLasso/Adjusted GWAS) for top DisGeNET common gene sets.
(PNG)

**S10 Fig. Comparison of DisGeNET Enrichment (Z-ratios) on DRIVE: MuGLasso vs. Adjusted GWAS.** On DRIVE, comparison of DisGeNET gene sets enrichment between MuGLasso and Adjusted GWAS based on Z-score ratios. Bar heights represent the ratio of Z-scores (MuGLasso/Adjusted GWAS) for top DisGeNET common gene sets.
(PNG)

**S1 Table. Subpopulations of *Arabidopsis thaliana* with the corresponding countries and the number of samples included in each subpopulation.**
(PDF)

**S2 Table. Potential breast cancer risk genes identified through physical (within 10 kb) mapping of the loci selected by Adjusted GWAS, SMuGLasso and MuGLasso.** CEU-specific selected genes are highlighted in blue and YRI-specific selected genes are highlighted in red. The remaining genes (in black) are risk genes shared across all populations.
(PDF)

**S3 Table. MuGLasso or/and SMuGLasso specific Genes in Meta-GWAS via Physical and eQTL Mapping.** Potential breast cancer risk genes identified through both physical (within 10 kb) and eQTL mapping of the loci selected by MuGLasso or/and SMuGLasso and not the adjusted GWAS, found in meta-GWAS including the samples used in this work.
(PDF)

**S4 Table. MuGLasso or/and SMuGLasso specific Genes linked to Breast Cancer in Literature.** Potential breast cancer risk genes identified through both physical (within 10 kb) and eQTL mapping of the loci selected by MuGLasso or/and SMuGLasso and not the adjusted GWAS, found to be associated with breast cancer risk or tumor growth in the literature.
(PDF)

**S5 Table. DTF3 loci detected by SMuGLasso and MuGLasso on *Arabidopsis thaliana* dataset**. Genes identified through physical mapping of SNPs selected as associated with flowering time in *Arabidopsis thaliana* using SMuGLasso, MuGLasso and Adjusted GWAS.
(PDF)

**S6 Table. Breast cancer risk loci detected by SMuGLasso and MuGLasso on DRIVE.** Potential breast cancer risk genes identified through both physical (within 10 kb) and eQTL mapping of the loci selected by Adjusted GWAS, SMuGLasso and MuGLasso. CEU-specific selected genes are highlighted in blue and YRI-specific selected genes are highlighted in red. The remaining genes (in black) are risk genes shared across all populations.
(PDF)

**S7 Table. MuGLasso or/and SMuGLasso specific eQTL Genes linked to Breast Cancer.** The potential breast cancer risk genes within 10 kb of loci obtained through eQTL analysis, identified by MuGLasso or/and SMuGLasso and not the adjusted GWAS, found to be associated with breast cancer risk or tumor growth in the literature.
(PDF)

**S8 Table. Stability index and number of selected features for different methods on *Arabidopsis thaliana*.**
(PDF)

**S9 Table. Summary of pathway and process enrichment analysis: Top 10 clusters of enriched terms, each described by one representative enriched term.** "#" is the number of genes in the user-provided lists with membership in the given ontology term. "%" is the percentage of genes selected by SMuGLasso that are found in the given ontology term (only input genes with at least one ontology term annotation are included in the calculation). "Log10(P)" is the p-value in log base 10. "Log10(q)" is the multi-test adjusted p-value in log base 10.
(PDF)

**S10 Table. Summary of enrichment analysis in DisGeNET.**
(PDF)

**S11 Table. Summary of enrichment analysis in Cell Type Signatures.**
(PDF)

**S12 Table. Summary of enrichment analysis in PaGenBase.**
(PDF)

**S13 Table. Summary of enrichment analysis in Transcription Factor Targets.**
(PDF)

## Acknowledgments

The authors would like to thank Adeline Fermanian, Vivien Goepp, Héctor Climente-González, Gwenaëlle Lemoine, Antoine Poirier and Lotfi Slim for fruitful discussion. OncoArray genotyping and phenotype data harmonization for the Discovery, Biology, and Risk of Inherited Variants in Breast Cancer (DRIVE) breast-cancer case control samples was supported by X01 HG007491 and U19 CA148065 and by Cancer Research UK (C1287/A16563).

## Author contributions

**Conceptualization:** Asma Nouira, Chloé-Agathe Azencott.

**Data curation:** Chloé-Agathe Azencott.

**Formal analysis:** Asma Nouira.

**Funding acquisition:** Chloé-Agathe Azencott.

**Investigation:** Asma Nouira, Chloé-Agathe Azencott.

**Methodology:** Asma Nouira, Chloé-Agathe Azencott.

**Project administration:** Chloé-Agathe Azencott.

**Resources:** Chloé-Agathe Azencott.

**Software:** Asma Nouira.

**Supervision:** Chloé-Agathe Azencott.

**Validation:** Chloé-Agathe Azencott.

**Visualization:** Asma Nouira, Chloé-Agathe Azencott.

**Writing – original draft:** Asma Nouira.

**Writing – review & editing:** Chloé-Agathe Azencott.

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
