## [Decision Letter · Decision Letter 0]

16 May 2025

PCOMPBIOL-D-24-02202

Sparse Multitask group Lasso for Genome-Wide Association Studies

PLOS Computational Biology

Dear Dr. Nouira,

Thank you for submitting your manuscript to PLOS Computational Biology. After careful consideration, we feel that it has merit but does not fully meet PLOS Computational Biology's publication criteria as it currently stands. Therefore, we invite you to submit a revised version of the manuscript that addresses the points raised during the review process.

Please submit your revised manuscript within 60 days Jul 16 2025 11:59PM. If you will need more time than this to complete your revisions, please reply to this message or contact the journal office at ploscompbiol@plos.org. Please include the following items when submitting your revised manuscript:

We look forward to receiving your revised manuscript.

Kind regards,

Adam Charles

Guest Editor

PLOS Computational Biology

Shihua Zhang

Section Editor

PLOS Computational Biology

**Additional Editor Comments:**

Dear Dr.'s Nouira and Agathe

Thank you for your patience in this review process. As you know it is increasingly difficult to solicit reviewers. Despite this we were able to receive sufficient reviewer feedback for your manuscript. Given the reviewer comments, the current manuscript lacks sufficient clarity, in particular int he details of the mathematical notation, selection of some of the model regularization, experimental results, and validation procedures to be published in its current form. However the reviewers did note the benefits of the approach and so I am suggesting a major revision, which should carefully address the specific reviewer feedback to improve the clarity of the manuscript.

Best,

-Adam

**Journal Requirements:**

At this stage, the following Authors/Authors require contributions: Asma Nouira, and Chloé-Agathe Azencott. Please ensure that the full contributions of each author are acknowledged in the "Add/Edit/Remove Authors" section of our submission form.

4) We notice that your supplementary Figures, and Tables are included in the manuscript file. Please remove them and upload them with the file type 'Supporting Information'. Please ensure that each Supporting Information file has a legend listed in the manuscript after the references list.

5) Please ensure that the funders and grant numbers match between the Financial Disclosure field and the Funding Information tab in your submission form. Note that the funders must be provided in the same order in both places as well.

**Reviewers' comments:**

Reviewer's Responses to Questions

**Comments to the Authors:**

Reviewer #1: In the manuscript, the authors extended the previous developed MuGLasso method to select population-specific risk variants with considering the selection of risk LD groups. The novel method, namely SMuGLasso, is based on a multitask group lasso framework. The manuscript is well-written, I have some major concerns related to the implementation details of the SMuGLasso:

1. The original MuGLasso requires additional post-processing steps to discern task-specific LD groups. What are the post-processing steps? And are they also be used for MuGLasso in the comparison experiments?

2. In the model, is X used to denote the raw genotypes of SNPs, or the normalized genotypes?

3. How to select hyperparameters lambda 1 and lambda 2?

4. How to pre-define the LD groups?

5. Are you using different variable selection methods on SNPs after LD pruning? How is the performance on SNPs without pruning?

6. Why different r2 thresholds are used for datasets in LD pruning?

7. In Figure 2, do runtimes of SMuGLasso and MuGLasso also incorporate the stability selection steps?

Minor issue:

Page 12, Line 363: For the Arabidopsis thaliana dataset (see ??)

Reviewer #2: I have uploaded my review in Word document, as it includes mathematical notations.

**Have the authors made all data and (if applicable) computational code underlying the findings in their manuscript fully available?**

Reviewer #1: Yes

Reviewer #2: **No: **At the time of my review, I did not have full access to the authors' simulation code or all simulation parameters. However, the real data components are clear.

PLOS authors have the option to publish the peer review history of their article (what does this mean?). If published, this will include your full peer review and any attached files.

Reviewer #1: No

Reviewer #2: **Yes: **Seungjun Ahn

**Figure resubmission:**
---

## [Decision Letter · Decision Letter 1]

27 Aug 2025

Dear Nouira,

We are pleased to inform you that your manuscript 'Sparse Multitask group Lasso for Genome-Wide Association Studies' has been provisionally accepted for publication in PLOS Computational Biology. The reviewer with the main concerns has re-reviewed the manuscript and found that the authors have answered their concerns and made the necessary modifications to the manuscript. While the other reviewer was unable to re-review, I have reviewed the responses and found that they address the main concerns. 

Best regards,

Adam Charles

Guest Editor

PLOS Computational Biology

Shihua Zhang

Section Editor

PLOS Computational Biology

Reviewer #2:

Reviewer's Responses to Questions

**Comments to the Authors:**

Reviewer #2: Thank you for your careful revisions. I find that you have addressed my previous comments and concerns, and I have no additional comments at this stage.

**Have the authors made all data and (if applicable) computational code underlying the findings in their manuscript fully available?**

Reviewer #2: None

PLOS authors have the option to publish the peer review history of their article (what does this mean?). If published, this will include your full peer review and any attached files.

Reviewer #2: **Yes: **Seungjun Ahn

---

## [Editor Report · Acceptance letter]

PCOMPBIOL-D-24-02202R1

Sparse Multitask group Lasso for Genome-Wide Association Studies

Dear Dr Nouira,

I am pleased to inform you that your manuscript has been formally accepted for publication in PLOS Computational Biology. Your manuscript is now with our production department and you will be notified of the publication date in due course.

With kind regards,

Zsofia Freund
